# Outcome and Subsequent Pregnancy after Fertility-Sparing Surgery of Early-Stage Cervical Cancers

**DOI:** 10.3390/ijerph17197103

**Published:** 2020-09-28

**Authors:** Chia-Yi Lee, Yu-Li Chen, Ying-Cheng Chiang, Ching-Yu Cheng, Yen-Ling Lai, Yi-Jou Tai, Heng-Cheng Hsu, Hsiao-Lin Hwa, Wen-Fang Cheng

**Affiliations:** 1Department of Obstetrics and Gynecology, College of Medicine, National Taiwan University, Taipei 100, Taiwan; plzfixthecar@gmail.com (C.-Y.L.); uly1007@yahoo.com.tw (Y.-L.C.); littlechiang1878@yahoo.com.tw (Y.-C.C.); b95401096@ntu.edu.tw (C.-Y.C.); adina@kimo.com (Y.-L.L.); stilabry@gmail.com (Y.-J.T.); b101092037@gmail.com (H.-C.H.); hwahl013@ntu.edu.tw (H.-L.H.); 2Department of Obstetrics and Gynecology, National Taiwan University Hospital, Yun-Lin Branch, Douliou City 640, Yunlin County, Taiwan; 3Department of Obstetrics and Gynecology, National Taiwan University Hospital, Hsin-Chu Branch, Hsin-Chu City 300, Taiwan; 4Graduate Institute of Clinical Medicine, College of Medicine, National Taiwan University, Taipei 100, Taiwan; 5Department and Graduate Institute of Forensic Medicine, College of Medicine, National Taiwan University, Taipei 100, Taiwan; 6Department of Medical Genetics, National Taiwan University Hospital, Taipei 100, Taiwan; 7Graduate Institute of Oncology, College of Medicine, National Taiwan University, Taipei 100, Taiwan

**Keywords:** cervical cancer, fertility-sparing surgery, conization, radical trachelectomy, pregnancy

## Abstract

We aimed to investigate the outcomes and subsequent pregnancies of early-stage cervical cancer patients who received conservative fertility-sparing surgery. Women with early-stage cervical cancer who underwent conservative or fertility-sparing surgery in a tertiary medical center were reviewed from 2004 to 2017. Each patient’s clinicopathologic characteristics, adjuvant therapy, subsequent pregnancy, and outcome were recorded. There were 32 women recruited, including 12 stage IA1 patients and 20 stage IB1 patients. Twenty-two patients received conization/LEEP and the other 10 patients received radical trachelectomy. Two patients did not complete the definite treatment after fertility-sparing surgery. There were 11 women who had subsequent pregnancies and nine had at least one live birth. The live birth rate was 73.3% (11/15). We conclude that patients with early-stage cervical cancer who undergo fertility-sparing surgery can have a successful pregnancy and delivery. However, patients must receive a detailed consultation before surgery and undergo definitive treatment, if indicated, and regular postoperative surveillance.

## 1. Introduction

Invasive cervical cancer (ICC) affects women of all ages. In 2016, 12,990 new cases of ICC were reported in the United States, and approximately 26.1% of them were in women younger than 40 years old [1]. In Taiwan, 1273 patients were diagnosed with ICC in 2015, and 422 of them (33.2%) were younger than 40 years old [2]. These patients were still of reproductive age, and some had not yet completed their family planning when diagnosed [3]. Loss of fertility caused by the treatment for malignancy can lead to grief, stress, sexual dysfunction, and depression in patients of reproductive age [4]. The development of treatment that preserves fertility, especially without compromising oncologic outcomes, is thus an important issue for these ICC patients.

The standard surgical treatment of early-stage ICC has traditionally been radical hysterectomy (RH) to cure the patient and prevent relapse of the disease [5]. Patients who receive this surgery have 80–87% 5-year disease-free survival and 84–87% 5-year overall survival [6,7]. However, RH sacrifices patients’ ability to get pregnant and brings complications including lower urinary tract dysfunction and sexual dysfunction [8,9]. Therefore, other less radical surgical methods for young ICC patients who still have fertility needs have been investigated [10,11,12]. For example, since Dargent et al. reported the use of laparoscopic radical vaginal trachelectomy with pelvic lymphadenectomy and patients’ subsequent pregnancies, radical trachelectomy has been used for patients with early-stage ICC and yielded satisfactory oncologic and pregnancy outcomes [13]. The recurrence rate is around 4% and the pregnancy rates are 50% to 80% among different surgical approaches of radical trachelectomy, including laparotomic, laparoscopic, or robotic-assisted radical trachelectomy [13,14]. For patients with stage IA2 disease (2009 FIGO staging system), conization or radical trachelectomy with bilateral pelvic lymph node dissection (BPLND) can be the standard treatment according to the ESMO Clinical Practice Guidelines of Cervical Cancer [15]. Patients with stage IB1 disease with a tumor <2 cm in diameter can also be considered for radical trachelectomy and BPLND [15]. On the other hand, FIGO stage IA1 disease without lymphovascular space invasion (LVSI) can be managed with more conservative treatment, including conization or simple trachelectomy [15]. Because there are few reports on the outcomes of patients with early-stage ICC treated with fertility-sparing surgeries (FSS) in Taiwan, we surveyed the ICC patients who were treated with FSS in our hospital and investigated their oncologic outcomes and subsequent pregnancies after surgery.

## 2. Materials and Methods

We retrospectively reviewed the medical records of patients diagnosed with ICC in one tertiary medical center from January 2004 to December 2017. The study was approved by the institutional review board of National Taiwan University Hospital (IRB: 201911041RINB). All of the patients’ data were fully anonymized before we accessed them and the Research Ethics Committee waived the requirement for informed consent. The inclusion criteria were as follows: (1) young age (<45 years old); (2) clinical stage IA1 to IB1 (2009 FIGO staging system); and (3) received FSS such as conization/LEEP (loop electrosurgical excision procedure), simple trachelectomy, or radical trachelectomy. Cold-knife conization (CKC) was the surgical method used when performing conization. The exclusion criteria included (1) age > 45 years; (2) previous subtotal hysterectomy history; (3) presence of rare histology other than squamous cell carcinoma, adenocarcinoma, and adenosquamous cell carcinoma, such as neuroendocrine or clear-cell carcinoma; and (4) received simple or radical hysterectomy less than 1 year after FSS. The clinical staging was achieved after patients received careful pelvic examination and colposcopy if necessary. Any suspicious lesion was biopsied and sent for pathologic review. Blood tests, chest X-ray, and electrocardiography were performed as pre-operative work-up. Abdomino-pelvic magnetic resonance imaging (MRI) was also arranged to assess the status of the parametria and lymph nodes before treatment.

Patients with microinvasive disease (stages IA1 and IA2) were diagnosed on the basis of their pathology results and treated by conization/LEEP. If the section margin was involved by invasive carcinoma but free of gross tumor, the disease was staged as stage IB1. If the patient received repeated conization, the pathological details of the repeated conization were recorded and counted as the final results determining the pathological stage. Pelvic lymph nodes were assessed on the basis of MRI before surgery. Patients with clinical IB1 disease received either conization, abdominal radical trachelectomy (aRT), or vaginal radical trachelectomy (vRT) plus BPLND, after a detailed discussion with the patients and their families. The indication and pros and cons of FSS were also thoroughly explained before surgery.

After primary surgery, the need for adjuvant therapy and the indicated modality were discussed by our institutional tumor board of gynecologic oncology. Periodic examinations, including history-taking, pelvic and rectal examinations, regional lymph node palpation, and pap smear, were arranged every 3 months for 2 years, and every 6 months for the subsequent 3 years. Any suspicious lesion noted during pelvic examination was biopsied and recurrence was defined after pathologic examination if available. Computerized tomography (CT) or MRI was arranged if disease recurrence was suspected, to detect local recurrence, pelvic lymphadenopathy, or distant metastasis. The disease-free survival (DFS) was defined as the time elapsed between the cancer diagnosis and the confirmation of recurrence or the date of the last out-patient clinic (OPD) visit without recurrence. Medical records after surgery were reviewed or telephone contact was recorded to assess subsequent pregnancy after FSS.

## 3. Results

A total of 32 patients with stage IA1–IB1 ICC who received FSS were recruited. The median age at diagnosis was 32 years (range, 24–42). Of these, fourteen women were unmarried, and 19 of them had not been pregnant. There were 12 patients with IA1 disease and 20 with IB1 disease. The median follow-up duration was 4.1 years (range, 1.4–15) (Table 1).

As shown in Table 2, among the 12 stage IA1 patients, the major histologic type was squamous cell carcinoma (SCC), with two patients having adenocarcinoma in situ. There were nine (75%) patients who received LEEP and three (25%) who received conization. Two patients had HSIL (high grade intraepithelial lesion) over the section margin of their LEEP or conization specimens. One patient had disease recurrence 38 months after LEEP. There were six (50%) patients who got pregnant after FSS, 18–40 months later. Four patients each had one live birth and one patient had two live births. One patient experienced a first trimester miscarriage due to a blighted ovum.

Among the 20 stage IB1 patients, there were 15 (75%) SCC and 5 (25%) adenocarcinomas. There were one and seven patients, respectively, who underwent conization and LEEP alone. Two patients underwent conization with BPLND, and the other 10 patients underwent aRT or vRT with BPLND. The median greatest tumor dimension was 13 mm (range, 1–32). Three patients had lesions over the surgical margin (one with HSIL and two with carcinoma). None of the 10 patients undergoing radical trachelectomy had parametrial invasion. One of 12 patients was found to have two pelvic lymph nodal metastases on pathological examination. Because the patient decided to preserve her ovarian function after consultation, adjuvant chemotherapy rather than chemoradiation was chosen based on previous report [16]. Two patients had vaginal invasion in the trachelectomic specimens. Three of these 20 patients had local disease relapse 5, 18, and 36 months after surgery, respectively. Five (25%) of the 20 women became pregnant after FSS, with a total of eight pregnancies. There were three women with one live birth and one woman with two live births. Two patients had a first trimester miscarriage and one patient received an artificial abortion due to unwanted pregnancy.

There were four women with disease relapse in this series including one HSIL and three ICC; none of them received definitive treatment after consultation. The characteristics of these four women are shown in Table 3. The first woman had stage IA1 disease and received LEEP alone, with an HSIL lesion over the exocervical surgical margin (Table 3, No. 1). She received follow-up rather than repeated LEEP after consultation. CIN3 lesion was proven by cervical biopsy 2 years later. The second woman had stage IB1 disease. She received diagnostic LEEP due to the uncertain invasive cervical carcinoma. The pathologic report of the LEEP revealed a 2.8 cm tumor in the greatest dimension with positive LVSI (Table 3, No. 2). Besides, the surgical margin of LEEP had CIN invovlement. There was no pelvic lymph node enlargement on MR images. However, she then refused definitive treatment after consultation. One 1-cm recurrent cervical SCC was diagnosed 18 months later. The third woman had stage IB1 disease and received only diagnostic LEEP, (Table 3, No. 3) with carcinoma involving the surgical margin. The patient refused definitive treatment even after consultation. She presented with a pelvic tumor combined with right-leg swelling and right hydronephrosis 3 years later. She then received salvage concurrent chemo-radiation (CCRT). The fourth recurrent patient had stage IB1 disease with a 3-cm cervical tumor. She underwent vRT and laparoscopic BPLND (Table 3, No. 4). Pathology reported vaginal invasion with positive LVSI but without surgical margin involvement. She refused adjuvant radiotherapy after consultation. However, intermittant flank pain and vaginal bleeding developed 5 months later. Pelvic examination revealed a 4-cm cervical tumor with vaginal invasion. Recurrent cervical SCC was dignosed. She then received salvage CCRT and lived without disease for more than 5 years.

The subsequent pregnancies of the 30 women (excluding two patients refused definitive treatment after fertility-sparing surgery) with early-stage ICC were further analyzed. As shown in Table 4, 11 (36.7%) of the 30 women were even pregnant after the FSS. The majority of the 11 women were treated with LEEP or conization (10/11, 90.9%). Only one woman was treated with aRT plus BPLND (Table 4, No. 9). Seven women had one live birth and two women had two live births after FSS. Two women experienced spontaneous abortion once before successful live birth. The median interval from the FSS to the first delivery was 2.7 years (1.5–4.6). Five women (5/9, 55.6%) received prophylactic cervical cerclage in the early second trimester. Of the 11 live births, nine (81.8%) were delivered by vaginal delivery and two (18.2%) by caesarean section. The preterm birth rate of these women was 54.5% (6/11). Five pregnancies were complicated by preterm premature rupture of membranes and these women were treated with intravenous antibiotics, tocolytics, and steroids in order to enhance fetal lung maturity. Two women had a first trimester miscarriage (Table 4, Nos. 10,11) without further pregnancy to date.

## 4. Discussion

Women with early-stage ICC had fair oncologic outcomes and subsequent pregnancies after FSS. Our results revealed that, firstly, LEEP/conization is feasible in stage IA1 ICC patients to preserve patients’ fertility. Secondly, FSS can be an option for stage IB1 disease with a small gross lesion (<2 cm). Conization provided better oncologic outcomes than LEEP. Thirdly, early-stage ICC patients who undergo FSS can have satisfactory pregnancy outcomes.

Early-stage ICC patients who receive conservative treatment have a risk of recurrence. Tomao et al. reported a 9% recurrence rate in women with early-stage ICC treated with conization [17]. Bentivegna et al. reported recurrence rates of 4% for vRT and 5% for aRT [14]. The recurrence rates were 13.6% for LEEP/conization, 0% for aRT, and 14.3% for vRT in this series. The pros and cons of FSS for early-stage ICC—including the chance for cure, the risk of recurrence, and the fertility results—should be discussed in full with patients [14]. The importance of surveillance after FSS should be also emphasized. In this series, the real recurrence rate was difficult to calculate, because two patients did not actually receive definitive treatment after FSS (Table 4), and some of the patients had poor compliance during postoperative surveillance. For patients who had surgical margin involvement in their LEEP specimens, we suggested repeated conization to completely excise the cervical lesion and to further evaluate the depth of cervical stromal invasion in the positive surgical margin of the first LEEP/conization surgery, as recommended by Ayhan et al. [18]. For patients with carcinoma involved surgical section margins, postoperative pelvic external beam radiation therapy with or without concurrent platinum-containing chemotherapy was recommended [19]. However, neither of the four relapsed patients underwent the above-mentioned treatments. Therefore, a patient’s good communication and compliance are important characteristics for being a candidate for FSS of early-stage ICC.

Patients with stage IA1 disease who desire to preserve their fertility are good candidates for LEEP or conization. Hartmann et al. reported a 7.3% recurrence rate in 41 IA1 patients treated with conization [20]. However, Shim et al. reported that there was no recurrence in a total of 93 patients who underwent conization or simple trachelectomy [21]. In our series, there was no invasive cancer recurrence in any of the stage IA1 patients. This oncologic outcome is comparable to those of previous studies.

Stage IB1 disease with tumors <2 cm can be a good indicator of conservative treatments for those who have not completed their family planning. Kato et al. retrospectively reviewed 604 stage IB1 patients and found that patients with tumors ≤2 cm had a low risk of parametrial involvement (1.9%) [22]. Kodama et al. also found that elderly ICC patients were more likely to have parametrial invasion, illustrating the importance of age [23]. Our previous study reported that only 1 of 109 patients younger than 50 years old who underwent radical hysterectomy had parametrial invasion [24]. The ongoing prospective trial CONCERV focused on the role of non-radical surgery for low risk early-stage cervical cancer [25]. The preliminary reported data showed only 3% recurrence in patients with stage IA2 to IB1 disease and tumors <2 cm in size [26]. In our series, none of the women with stage IB1 tumors <2 cm in size had disease recurrence. The standard management of these women is still radical trachelectomy. We recommend, for women with tumor size <2 cm who desire to retain their fertility, that less radical surgery can be considered with in depth patient counseling or as part of a clinical trial.

Stage IB1 diseases treated with LEEP have a higher risk of recurrence than those treated with conization. Conization, especially CKC, can remove the ectocervical and endocervical canals en bloc and spare electrosurgical artifacts [18]. Miroshnichenko et al. observed that LEEP specimens have limited interpretability and are more likely to have positive margins [27]. They suggested CKC or conization rather than LEEP to achieve a more complete resection for stage IB1 disease. None of the three women with stage IB1 disease treated with conization had disease recurrence in our series. Therefore, we recommend conization, especially CKC, and not LEEP for fertility-sparing treatment of stage IB1 disease.

The tumor size of early-stage ICC might be under-estimated or measured. According to international guidelines [15], radical trachelectomy is the standard surgical procedure for FSS of women with stage IB1 disease and tumor size <2 cm, since tumors of size >2 cm frequently require postoperative adjuvant therapy due to pathologic risk factors such as deep stromal invasion or LVSI [28]. In this study, of the 10 women with stage IB1 disease treated with radical trachelectomy, two had tumor sizes >2 cm in their surgical pathologic specimens. One of them who did not receive adjuvant therapy had disease recurrence 5 months later. The other received adjuvant chemotherapy after surgery and was disease free for 77 months. Whether radical trachelectomy is feasible for tumors >2 cm in size is still debated. Lintner et al. reported that radical trachelectomy and radical hysterectomy had similar 5-year survival results in patients whose cervical tumors were between 2 and 4 cm in size [29]. Li et al. suggested the upper limits of tumor size are 2 cm for vRT and 4 cm for aRT [28]. MR imaging has been recommended for assessing primary cervical tumors of >10 mm [30]. Moreover, 18F-fluorodeoxyglucose-positron emission tomography (FDG-PET) tumor size measurements correlate highly with pathological tumor size. PET-CT or PET-MR also provide more accurate evaluation of lymph node metastases [31]. Therefore, we recommend using a comprehensive preoperative imaging study in choosing the optimal surgical procedure for any woman with early-stage ICC who is preparing for FSS.

Minimally invasive approaches have been used in RT. Laparoscopic RT was first reported in 2000 [13], and robotic RT was first described in 2008 [32]. The advantages of minimally invasive surgery in RT include shorter length of hospital stay, reduced blood loss, decreased analgesic requirements, decrease in blood transfusion rates, decrease in complication rates, and earlier recovery of physiological functions [33]. Compared to aRT, patients who underwent either laparoscopic or robotic RT had significantly better short-term wellbeing [34]. However, there were limited data on the safety and long-term oncological outcomes.

Early-stage ICC patients could get pregnant and bear children after conservative treatment or FSS. All of our patients expressed a desire for future pregnancies during consultation and therefore underwent FSS. In total, 11 of 30 women had one or more pregnancies after FSS, including 10 of 20 women treated with conization or LEEP and 1 of 10 women treated with radical trachelectomy. Nine (81.8%) of 11 women had live births, including two who had two live births. The total number of pregnancies was 15, as shown in Table 4. The live birth rate of our series was 73.3% (11/15), similar to other studies. Linsay et al. reported a live birth rate of 42% in their 43 early-stage ICC patients treated with LEEP [35]. Okugawa et al. reported 15 live births out of a total 21 pregnancies (71.4%) in cases treated with aRT [36]. Therefore, child-bearing can be achieved by selected early-stage ICC women after undergoing FSS. Preterm birth was a frequent complication in the ICC patients who underwent FSS [37,38]. One explanation for this is subclinical or clinical chorioamnionitis caused by a shortened cervix and the potential exposure of the amniotic sac to the vagina [38]. Bentivegna et al. reported prematurity rates of 15% in conization/simple trachelectomy, 39% in vRT, and 57% in aRT [38]. Premature preterm rupture of membranes (PPROM) was a cause of second trimester fetal losses [38]. The prematurity rate was 54.5% in this series, including five pregnancies subsequent to LEEP/conization complicated with PPROM even when treated with tocolytic agents. The pregnancy outcomes of the LEEP/conization group were better than those of the radical trachelectomy group, due to the less extensive parametrial excision and milder damage to the pelvic floor, which can reduce the pregnancy loss rate [39]. With trachelectomy, more favorable pregnancy outcomes have been reported when patients received cervical cerclage [40]. However, there is no consensus regarding the most appropriate method and type of cerclage [40]. Although there were two women in our study who had term delivery without prior cervical cerclage, we still recommend prophylactic cervical cerclage for these women.

## 5. Conclusions

Fertility-preserving surgeries, including LEEP, conization, and radical trachelectomy, can be safe alternatives to hysterectomy in selected young patients with early-stage cervical cancer who desire to retain their fertility. We recommend a detailed consultation before surgery, undergoing definitive treatment if indicated, and regular surveillance of these patients after FSS.

## Figures and Tables

**Table 1 ijerph-17-07103-t001:** Basic characteristics of 32 early-stage invasive cervical cancer patients treated with fertility-sparing surgery.

Characteristics	Number of Patients (%)
Age (y/o) (median, range)	32 (24–42)
Marital status	
No	14 (43.8)
Yes	18 (56.2)
Parity	
Nulliparity	19 (59.4)
Multiparity	13 (40.6)
FIGO stage *	
IA1	12 (37.5)
IB1	20 (62.5)
Follow-up (years) (median, range)	4.1 (1.4–15)

y/o: year old, * 2009 FIGO staging system.

**Table 2 ijerph-17-07103-t002:** Characteristics of 12 stage IA1 and 20 stage IB1 invasive cervical cancer patients treated with fertility-sparing surgery.

Characteristics	Number of Patients (%)	
**Stage**	IA1	IB1
Histology		
SCC	10 (83.3)	15 (75)
SCC with AIS	2 (16.7)	0 (0)
Adenocarcinoma	0 (0)	5 (25)
Treatment		
LEEP	9 (75)	7 (35)
Conization	3 (25)	1 (5)
Conization + PLND	0 (0)	2 (10)
Abdominal RT + PLND	0 (0)	3 (15)
Vaginal RT + PLND	0(0)	7 (35)
Tumor greatest dimension (mm) (median, range)	4 (2–6)	13 (1–32)
Depth of stromal invasion (mm)(median, range)	1.75 (1–2)	N/A
Surgical margin involvement		
No	10 (83.3)	17 (85)
HSIL	2 (16.7)	1 (5)
Carcinoma	0 (0)	2 (10)
Parametrial involvement		
No		10 (50)
N/A	12 (100)	10 (50)
Vaginal involvement		
No		18 (90)
Yes		2 (10)
N/A	12 (100)	0 (0)
LN involvement *		
No		14 (70)
Yes		1 (5)
N/A	12 (100)	5 (25)
LVSI		
No	12 (100)	4 (20)
Yes	0 (0)	16 (80)
Recurrence		
No	11 (90.9)	17 (85)
Yes	1 (9.1) ^+^	3 (15) ^++^
Subsequent pregnancy		
Yes	6 (50)	5 (25)
Abortion	1 (8.3)	1 (5)
Delivery		
1	4 (33.3)	3 (15)
≥2	1 (8.3)	1 (5)
No	6 (50)	15 (75)

SCC: squamous cell carcinoma; AIS: adenocarcinoma in situ; LEEP: loop electrosurgical excision procedure; PLND: pelvic lymph node dissection; RT: radical trachelectomy; HSIL: high-grade squamous intraepithelial lesion; N/A: not applicable; LN: lymph node; LVSI: lymphovascular invasion; * assessment of LN status: either by preoperative MRI or pathologic examination; ^+^ recurrent CIN3; **^++^**: two of them did not receive definite treatment.

**Table 3 ijerph-17-07103-t003:** Characteristics of four early-stage invasive cervical cancer women with recurrent CIN or ICC without definitive treatment after fertility-sparing surgery.

No.	1	2	3	4
Age (y)	37	32	30	29
Parity	1	0	0	0
Clinical stage *	IA1	IB1	IB1	IB1
Pathologic stage *	IA1	IB1	IB1	IIA1
Management	LEEP	LEEP	LEEP	vRT with LSC BPLND
Histology	SCC, AIS	SCC	SCC	SCC
Tumor largest dimension (mm)	1	28	6	30
Stromal invasive depth (mm)	1	10	5	5
LVSI	No	Yes	No	Yes
LN	N/A	No **	N/A	No
Parametrial invasion	N/A	N/A	N/A	No
Vaginal invasion	N/A	N/A	N/A	Yes
Surgical margin involvement	Yes (HSIL)	Yes (HSIL)	Yes (carcinoma)	No
DFS (months)	24	18	36	5

CIN: cervical intraepithelial neoplasia, ICC, invasive cervical carcinoma; * FIGO 2009 stage guideline; LEEP, loop electrosurgical excision procedure; vRT, vaginal radical trachelectomy; LSC PLND, laparoscopic pelvic lymph node dissection; SCC, squamous cell carcinoma; AIS, adenocarcinoma in situ; LVSI, lymphovascular invasion; LN, lymph node; DFS, disease-free survival; N/A, not available; HSIL, high-grade squamous intraepithelial lesion; ** assessment of LN status by MRI.

**Table 4 ijerph-17-07103-t004:** Characteristics of the subsequent pregnancies of 11 early-stage invasive cervical cancer patients treated with fertility-sparing surgery.

No.	Age (y)	Clinical Stage	Pathologic Stage	Surgical Procedure	Subsequent Gravida	Subsequent Parity	Interval (y)	Cerclage	Cervical Incompetence	Remarks
**1**	29	IA1	IA1	LEEP	1	1	1.5	Yes	Yes	Term, NSD
**2**	32	IA1	IA1	LEEP	1	1	1.7	No	No	Term, NSD
**3**	37	IA1	IA1	Conization	1	1	3.3	No	No	Term, NSD
**4**	29	IA1	IA1	LEEP	2	2	2.7	No	No	GA 33^+4^ weeks, GA 35^+5^ weeks, both NSD
**5**	29	IA1	IA1	LEEP	1	1	3.1	No	No	GA 36^+4^ weeks, C/S
**6**	33	IB1	IB1	Conization with LSC BPLND	1	1	4.4	Yes	No	GA 32^+6^ weeks, C/S
**7**	31	IB1	IB1	LEEP	2	1	4.6	Yes	No	GA 36^+0^ weeks, NSD
**8**	38	IB1	IB1	LEEP	2	1	1.5	Yes	No	GA 34^+5^ weeks, NSD
**9**	28	IB1	IIA1	aRT with BPLND	2	2	2.4	Yes	No	Adjuvant chemotherapy Term, NSD
**10**	41	IA1	IA1	LEEP	1	0	2.4	N/A	N/A	Blighted ovum
**11**	30	IB1	IB1	LEEP	1	0	2.5	N/A	N/A	Unwanted pregnancy

LEEP, loop electrosurgical excision procedure; NSD, normal spontaneous delivery; PPROM, preterm premature rupture of membrane; C/S, cesarean section; LSC BPLND, laparoscopic bilateral pelvic lymph node dissection; aRT, abdominal radical trachelectomy; N/A, not applicable, GA: gestational age.

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
