# Peer review of "Outcome and Subsequent Pregnancy after Fertility-Sparing Surgery of Early-Stage Cervical Cancers"

_ijerph, 2020, doi:10.3390/ijerph17197103_

Round 1
Reviewer 1 Report
This is a small retrospective series looking at fertility outcomes following fertility-sparing surgery (FSS) for invasive cervical cancers in Taiwan. Their reported outcomes are similar to other series, but their overall study number (32) is very low to draw strong conclusions. The majority of patients in this manuscript were treated with LEEP/conization so I might have expected a higher pregnancy rate overall.
Since the authors spent a lot of time reporting and discussing patient outcomes, they may want to consider broadening the objective of the paper to include both oncologic and fertility outcomes for FSS.
The only other thing I would consider rewording is the recommendation for less radical surgery (i.e. conization) for IB1 tumors. This is still an investigational approach in the United States (NCCN guidelines recommend radical trachelectomy still) but it can be considered with in depth patient counselling or if part of a clinical trial.
Reviewer 2 Report
This is a well written paper on a very pertinent issue, however it has many drawbacks.
When we seek information about the outcome and subsequent pregnancy after fertility sparing surgery of early-stage cervical cancers, we try to know whether a conservative treatment in well selected patients may simultaneously treat (hopefully cure) properly the patient, and preserve her fertility. That conservative treatment is usually a patient´s informed choice, after other standard options adequate to the stage are presented to her, because she wants or considers to become pregnant one day.
Nevertheless, what we see here is a number of cases in which a fertility sparing treatment was performed, not in the context described above, but rather as a result of under/mistreatment for reasons that are not always completely clear.
For instance:
==(Page 5, line 129) «The second recurrent case had stage IB1 disease, with a 2.8 cm cervical tumor (Table 4, No. 2). She underwent LEEP alone without definitive treatment planning, for personal reasons. There was no pelvic lymph node enlargement on MR images. The surgical margin of LEEP still had HSIL.»
-Why, in first place, a patient with a 2.8 cm cervical tumor underwent a LEEP? She was not even a good candidate for tracheletomy..And a diagnostic LEEP would not be necessary in a tumor of this size.
==(Page 5, line133) «The third woman with recurrence had stage IB1 disease. She received only diagnostic LEEP, (Table 4, No. 3) with carcinoma involving the surgical margin. The patient refused definitive treatment.»
-Regardless of the reasons behind the patient´s decision, she was overtly undertreated. Therefore, this case cannot also be taken into account when considering the recurrence rate/outcome of fertility sparing treatment.
==(Page 5, line 136) «The fourth recurrent patient had stage IB1 disease with a 3-cm cervical tumor. She underwent vRT and laparoscopic BPLND..»
-Again, this is not the ideal candidate for trachelectomy.
==(Page 4, line 108) The large proportion of IB1 patients that underwent only LEEP/conization.
==(Pag 4, line 113) «One of 12 patients was found to have pelvic lymph node metastasis, when undergoing BPLND.»
-The presence of lymph node metastasis is a contraindication for fertility sparing treatment, as it requires chemoradiation. Did the patient receive chemoradiation? If the answer is yes, how can she be included in the study? If it is no, again she was undertreated.
-When there is indication for BPLND, the institution does not use to check the lymph node status before proceeding to a fertility sparing treatment?
There are also other questions:
==(Page5, line 127) «The first recurrent case had stage IA1 disease and received LEEP alone, with an HSIL lesion over the exocervical surgical margin (Table 4, No. 1). She received follow-up rather than repeated LEEP. Recurrent HSIL was proven by cervical biopsy 2 years after LEEP.»
-It cannot be considered recurrence, as it was not an invasive carcinoma.
==It is said that patients underwent conization or LEEP. However, conization can be performed with a scalpel (cold-knife conization), laser, or electrosurgical loop. The latter may also be called the loop electrosurgical excision procedure (LEEP). Therefore, it is not clear why these two groups were displayed, particularly because the conization method(s) was(were) not disclosed.
==It would be better to clarify which FIGO staging system was used (2009 or 2018) right in the Introduction, and not only in Methods, as when we start reading the paper it is not clear and confusion may rise.
==Did all patients in which a pregnancy did not occur attempted a pregnancy? We do not know for sure the probability of pregnancy after fertility sparing treatment if we do not know if a pregnancy was attempted or not.
==An effort should be made to mix data from Tables 2 and 3 in a single table.
Reviewer 3 Report
Congratulation to the authors . I've read the results of the retrospective studies with great interest and I aprreciate in depth analysis and practical value of this work.
To additional comments in manuscript:
-About assisted reproductive technology (how many women required this procedure, and their pregnancy experience
-minimally invasive approaches to fertility-sparing surgery ( in report luck of these data)
-and add the review literature about this topic.
